# Are Jobs Available in the Market? A Perspective from the Supply Side

**Salwaty Jamaludin, Rusmawati Said \*, Normaz Wana Ismail and Norashidah Mohamed Nor**

School of Business and Economics UPM, Universiti Putra Malaysia, Serdang 43400, Selangor Darul Ehsan, Malaysia; salwaty.jamaludin@gmail.com (S.J.); nwi@upm.edu.my (N.W.I.); norashidah@upm.edu.my (N.M.N.)
\* Correspondence: rusmawati@upm.edu.my

**Abstract:** Graduate unemployment exhibits a clear increasing global trend, and Malaysia is no exception. The unemployment rate among graduates is witnessing a considerable upsurge, growing from 43,800 in 2000 (15% of total unemployed) to more than 175,500 in 2017 (35%). Numerous programmes have been implemented in order to secure jobs for the unemployed in the labour market; however, the number of unemployed graduates keeps on increasing. It is significant to recognise the main reason behind this issue to tackle the risk of long-term unemployment, specifically from the supply side. Using the Relative Importance Index (RII), this study investigated 402 respondents at selected job fairs to identify the cause of their difficulty in entering the labour market. The findings revealed that the unemployed people believe that the principal cause of their unemployment is the lack of suitable jobs for them in the market. This circumstance sends a signal of asymmetric information between demand and supply in the labour market, especially to young graduates.

**Keywords:** unemployment; mismatch; vacancies





## 1. Introduction

Over the period, more than half a million secondary students enrol in tertiary education institutions as an investment in education which promises better future career prospects. In Malaysia, there were 581,668 students enrolled at university in 2015, 540,638 students in 2016, 532,049 students in 2017, 538,555 students in 2018, and 552,702 students in 2019, with the popular courses taken being Social Sciences; Business and Law; Engineering; Manufacturing and Construction; and Science, Mathematics, and Computing.

After a few years of studying, graduates expect to enter the formal labour market with an attractive salary. Yet, it is not as easy as this. The unemployment rates of these groups have increased remarkably, even though the national unemployment rate has hovered just around 3.3% in recent years. In 2000, the proportion of unemployed people with tertiary education was only 15%; however, in 17 years, the number has reached 35%, as shown in Figure 1.

The unemployment rate has worsened due to the COVID-19 pandemic—it reached a peak of 5.3% as of May 2020. Recession has happed everywhere. The pandemic has caused employers to cut their losses by cutting hundreds of thousands of jobs as the pandemic continues to affect the economy. The erosion of human capital has accumulated due to the loss of both schooling and jobs. There have been 2.7 million own-account workers who are at risk of unemployment, followed by the slow growth of filled-in jobs and decrease in skilled job vacancies [1].

Meanwhile, from the demand side, even though millions of job vacancies were recorded, low-skilled vacancies are the most representative. Low-skilled vacancies made up an average of 65% of total vacancies from 2006 to 2017, as shown in Figure 2. During the same period, on average, medium-skilled vacancies accounted for 26%, while high-skilled vacancies accounted for only 9%.

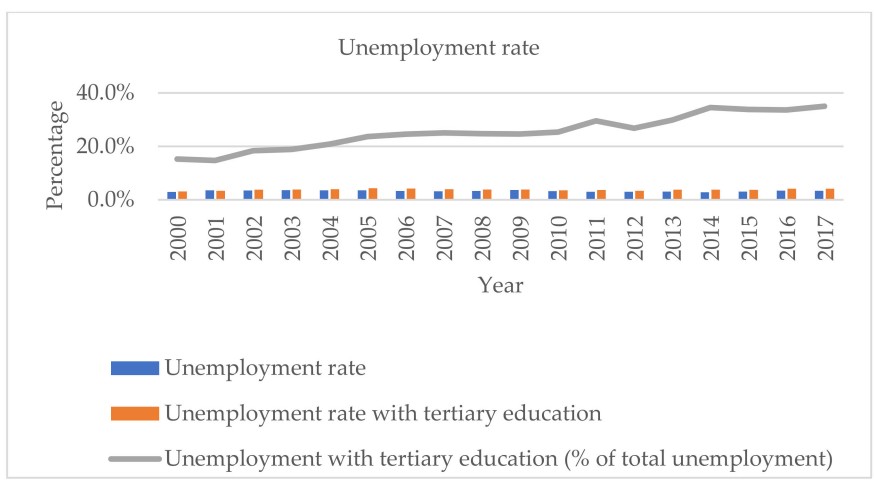

**Figure 1.** Unemployment rate in Malaysia. Source: Department of Statistics, Malaysia (DOSM).

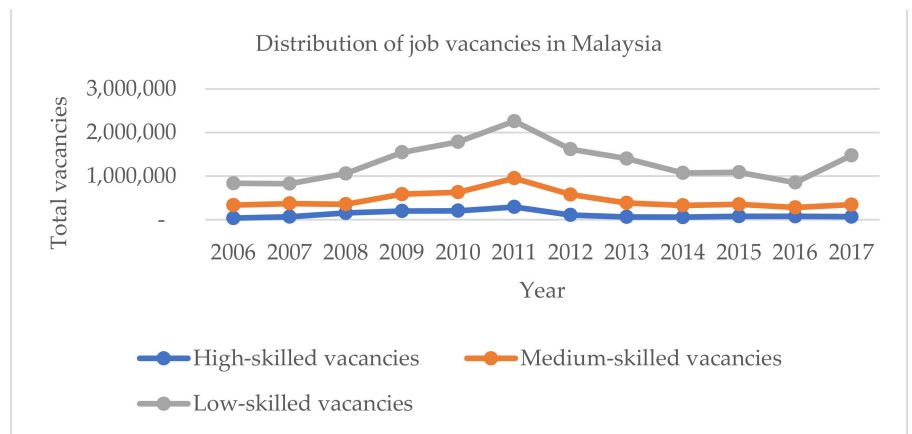

**Figure 2.** Job vacancy distribution in Malaysia based on skills. Sources: Ministry of Human Resource (MOHR) and Bank Negara Malaysia (BNM).

The highest number of vacancies was recorded in 2011, with approximately 2.3 million vacancies. The services sector contributed the most vacancies (34%), followed by the manufacturing (31%), agriculture, forestry, and fishing (18%), and construction (17%) sectors. In the services sector, the subsectors of finance, insurance, real estate, and business activities contributed significantly to the total (16%). From 2014 to 2015, the number of vacancies shrank by almost 50%. On average, manufacturing contributed to 31% of the total, services 26%, agriculture, forestry, and fishing 24%, and construction 19%. From the services sector, the wholesale and retail trade and hotels and restaurants subsectors contributed the most to the total vacancies (9%). In 2017, there were almost 1.5 million vacancies.

To disentangle the issue of unemployment as a whole, tracing the reasons behind it is important. From the demand side, it is reported that recent graduates lack certain so-called "soft skills" required by employers. For example, they have poor communication skills. Another reason is that the applicants do not meet the required skills or experience criteria. In some cases, applicants are qualified for the job, but employers are unable to meet the requested salary. Other studies have highlighted that graduate unemployment in Malaysia is caused by the not-work-ready mentality of the graduates, and not due to the lack of job opportunities.

Most primary studies have focused on the demand-side factors and thus motivated us to carry out systematic primary research to allow for better understanding of supply-side factors that delay the unemployed from entering the formal labour market. As the

preferences of job seekers and employers in the matching process have not been much explored, despite increasing interest in the field, this issue will be addressed in this study.

A different approach was conducted with a micro-level analysis by investigating 402 unemployed graduates to ascertain the principal reason for their unemployment. Following this introduction, the article briefly discusses the state of graduate unemployment in Malaysia and includes a literature review with the methodology. The next part reports the initial findings and a discussion of the reasons for being unemployed. The final section draws conclusions from the results of the analysis and suggests measures to be undertaken to improve the current labour market situation.

## 2. Literature Review

### 2.1. Theoretical Review

Human capital can be defined as the skills, knowledge, and experiences of an individual, employers, or a country as a whole that can generate economic value. In [2], human capital investment is defined as any form of investment in education made by an individual for future returns that are expected to exceed the current costs of participating in education. Returns are of two kinds: earnings premium for the individual and higher productivity for the firm.

Since an identical level of human capital can be replicated by the combination of education, training, and experience—that is, a trade-off between education and other forms of human capital—workers may qualify for similar jobs while having different levels of schooling [3]. However, this theory plays no role in the demand side, as it posits that economic growth solely depends on the accumulation of human capital from the supply side. The hypothesis is that more years of education correlate to higher earnings. The paper [4] is among the studies that applies human capital together with the Job Competition Model and assignment theories to frame the issue of overqualification and the relationship to wages.

### 2.2. Empirical Review

A considerable amount of factors that contribute to unemployment at a macro level have been well documented. For European countries [5], it has been revealed that German Foreign Direct Investment (FDI) inflows increase economic growth rate and boost employment in various economic sectors. However, for the case of Malaysia, even though Malaysia is a major recipient of FDIs, which has created many jobs for the locals, yet [6] it has been shown to correlate negatively with skilled labour demand. In terms of trade, [7] suggests that trade balance will increase employment opportunities for both skilled and unskilled labour in Malaysia. Thus, it could become a catalyst to reduce Malaysia's unemployment rate in the future.

At the micro level, from the demand side, the general opinion among Malaysian employers suggests that Malaysian graduates are well qualified in their fields of specialisation but, unfortunately, lack "soft skills" [8]. This "deficit" in graduate skills has also been acknowledged by [9,10]. The study suggests that the shortcomings of Malaysian graduates include the lack of a strong command of English, leadership, and other professional skills. These factors contribute to difficulties for graduates in finding jobs that suit their requirements.

Similarly, [11] reports that the increase in graduate unemployment is mainly due to the mismatch between the skills of graduates and those required by employers. The issue has also been highlighted in the Malaysia Plan, which maintains that the main problem in recent years has not been in the supply of graduates but in the skills that the graduates lack, such as leadership, decision-making, and communication skills.

Furthermore, [12] found that one reason adding to the unemployment issue among Malaysian graduates is the quality of graduates. There are employers in the sector who have made critical remarks on graduates and have suggested that graduates do not have the requisite expertise and credentials to fulfil the needs of industry. In comparison, learners

are poor in terms of employability capabilities and do not show good job performance. This is also supported by [13], which highlighted the same finding, with two additional factors—the expertise of graduates and the expertise of lecturers, which are among the factors that lead to the unemployment problem among graduates in technical fields today.

In addition, according to [14], another factor was suggested that contributes to unemployment. Aside from the lack of proficiency in English, the discrepancy between the form of graduate degree and the criteria for available employment in the labour market, the nature of the job, self-expectation on employability, the state of satisfaction in overall life, and the family context play significant roles.

Furthermore, [15] suggests that, other significant factors contributing to the unemployment issue among graduates should be highlighted. These include the rapid population growth, rapid decline in mortality rates, rapid growth of a graduate labour force, educational advancement, and the lack of relationship between educational institutions and industry.

## 3. Methodology

### 3.1. Questionnaire Development

A survey questionnaire was developed that consists of two sections and attempts to distinguish job availability choices as determinants of unemployment, especially for unemployed graduates.

The first section describes demographic information such as age, gender, marital status, location, education level, education background, and year of graduation. Based on [16], researchers often collect demographic information in research surveys for two purposes. The first is to answer their research questions if these involve demographic information as independent variables. The second reason is to collect demographic information which accurately describes their sample. The second part of this study sought feedback on graduates' difficulties in entering the labour market. The full questionnaire is shown in Table 1.

**Table 1.** List of questions.

| Section | Question |
|---------|----------|
| Section A: | Age<br>Gender<br>Marital status<br>Current residence<br>Education level<br>Education background<br>Year of graduation |
| Section B: | (1) How long you have been looking for a job?<br>(2) On average, how often do you take the initiative to find a job?<br>(3) How do you find the information about a job vacancy? Rank according to the level of importance.<br>(4) Why is it difficult for you to find a job? Rank the factors according to the level of importance.<br>(5) What are your criteria to accept a job offer? Rank them according to the level of importance.<br>(6) Please rank the sector you wish to venture in based on your priorities.<br>(7) Do you mind working in a sector that is unrelated to your background and level of education? Give your reason. |

Some of the questions from the Section B were adapted from the other countries' Labour Force Survey (LFS). The LFS questionnaires in other countries such as Japan, the United Kingdom and Uganda have highlighted the attribute of "No jobs available" as a reason for unemployment or reason for not looking for work, as shown in Table 2:

**Table 2.** Other countries' Labour Force Survey (LFS) questionnaires.

| Country | Japan | United Kingdom | Uganda |
|---|---|---|---|
| Types of questions | What is the reason you do not get a job?<br><br>• Not satisfied with the salary available job<br>• Unfavourable working hours and days<br>• Limited age<br>• Need more skills or knowledge<br>• Preferable kind of jobs not available.<br>• Wish to have any kinds of job, but not available<br>• Other | What is the reason you did not look for work?<br><br>• Waiting for the results of an application for a job/being assessed by a training agent<br>• Student<br>• Looking after the family/home<br>• Temporarily sick or injured<br>• Long-term sick or disabled<br>• Believe(s) no jobs available<br>• Not yet started looking<br>• Do(does) not need employment<br>• Retired from paid work<br>• Any other reason | What is the main reason you did not try to find work or start a business?<br><br>• Was waiting for the results of an interview<br>• Awaiting the season for work<br>• Attended school or training courses<br>• Family responsibilities or housework<br>• Pregnancy<br>• Illness, injury or disability<br>• Does not know how and where to look for work<br>• Unable to find work for his/her skills<br>• Had looked for job(s) before but had not found any<br>• Too young or too old to find a job<br>• No jobs available in the area/district<br>• Other reason |

Source: Japan's LFS, UK's LFS and Uganda's LFS.

The attribute of "No jobs available" was not emphasised in Malaysia's LFS questionnaire, either to an active job seekers or passive job seekers. Thus, it was necessary to include this attribute, as it could be a reason for unemployment. The selection of other attributes is explained in Table 3.

**Table 3.** Factors of unemployment.

| Item | Factor | Explanation | Source |
|---|---|---|---|
| 1. | No available jobs | Characteristics of jobseekers match poorly with the characteristics of vacancies. | [17] |
| | | Asymmetric information in the labour market could lead to mismatch. | [18] |
| | | Lack of information about the transition between university and the job market. | [19] |
| 2. | Wages | Rigid wages as one of the explanations for variations in unemployment. | [20] |
| | | Negative effect of education–job mismatch on wages. | [21] |

**Table 3.** *Cont.*

| Item | Factor | Explanation | Source |
|---|---|---|---|
| 3. | Educational mismatch | Likelihood of being mismatched rises with the level of highest attained education. | [22,23] |
| | | Technological shock. | [24,25] |
| 4. | Soft skills mismatch | Soft skills do not match employer's demands. | [24,25] |
| | | Technological shock. | |
| 5. | Location mismatch | The unemployed are not situated in regions where employment is growing. | [26] |
| | | Jobseekers may be qualified for the job, but they have limited access to reach the employment centres. Location of job concentration is too far from their place of residence. | [27] |
| 6. | Family responsibilities | For married couples, the search for jobs will be constrained geographically, and rarely will both spouses get the best job offers in the same location. As the husband's job choice is more dominant because of the wife's home responsibilities, married women will face greater constraints, whether as tied-stayers or tied-movers. | [28] |
| 7. | Others | Other factors that may contribute to unemployment. | |

These questions provide a small but potentially important insight into individual preferences in terms of the role of each variable in influencing the decision to enter the labour market. Next, for the second question, in order to check the consistency of the results from the question above, the factors which are important when considering a job offer were added.

The question was as follows:

What are your priority criteria when choosing to accept a job offer? Rank according to the level of importance.

The selection of attributes was as follows:

- Qualification;
- Wages and benefits;
- Location;
- Job satisfaction;
- Firm/organisation culture;
- Family responsibilities;
- Others.

Five out of seven items are similar to the first question, while another two items were added, job satisfaction and organisation culture, as these two items could influence jobseekers' entry to the labour market. A study conducted by [29], which studied the job preferences of Generation Y, revealed that this generation prefers job security and a relaxed work atmosphere. These two elements are among the most important elements of job satisfaction. Specifically, job satisfaction could be defined as the extent to which an employee feels self-motivated, pleased, and satisfied with his/her job. It also happens when an employee feels he or she has job stability, career growth, and a comfortable work–life balance. Several studies have measured the influence of educational mismatch and skills mismatch on job satisfaction. Most studies found that being a mismatched worker, specifically one who is overqualified, could adversely affect job satisfaction [30–32]. Thus, it is an important element to be considered when the decision is taken to engage with the labour market.

Firm or organisation culture also could influence an individual's decision to accept a job offer. Based on [33], work culture in an organisation could be defined as social and be-havioural expectations in the form of both informal and formal interactions. These include expectations regarding dress, interaction, and topics of discussion, whether informally in the lunchroom or in more formal work groups. Potential employees might search for this

kind of information before they make a decision to take up employment with the company. The study [34] suggested that jobseekers could increase their future job satisfaction by selecting organisations with cultures that match their values, as an expert advised jobseekers to consider the values of recruiting organisations in their career decisions.

### 3.2. Choice of Sampling

A total of 402 responses were gathered from a stratified random sampling of unemployed graduates at job fairs organised by JobsMalaysia and the Malaysian Investment Development Authority (MIDA) in order to concentrate on the population of active job seekers. Since a job fair is an event at which employers, recruiters, and schools give information to potential employees, attending job fairs would represent a great opportunity for the unemployed in their job search. In addition, a job fair is significant for encouraging those who are looking for jobs to assert themselves in the formal labour market, and also for them to exchange information about labour market prospects [35]. Career fairs could also broaden students' career interests and enhance their sense of self-efficacy [36].

The determination of the sample size followed [37], with a 5% margin of error, which led to estimating that the minimum number of respondents would be 383 after taking into consideration the total number of unemployed graduates in Malaysia, which had reached 152,100 graduates in 2015. The data used in this study were collected between February 2017 and September 2017 in the southern region (Johor), northern region (Penang), western region (Selangor), and eastern region (Terengganu) of Malaysia.

The strategic location of Johor, which is nearest to Singapore, was chosen among other states in the southern area. The huge wage gap between Malaysia and Singapore causes citizens in Johor to commute to Singapore every morning, even though there are many job vacancies in Johor, which had around 250,453 jobs in 2017 [38]. The migration of highly qualified graduates that could not meet the demand in their own country also disturbs the flow of demand and supply in the job market itself.

Meanwhile, in the northern region, job vacancies are concentrated in the Penang area, which has 57,737 jobs compared to Kedah and Perlis, which created only 18,650 and 1045 jobs, respectively [39]. Penang benefits from an advanced manufacturing hub, which focuses on electric and electronic products that contribute to high GDP growth. For the western region, Selangor was chosen as it is the closest state to the capital of Malaysia, Kuala Lumpur. It is an employment concentration area, with total job vacancies of up to 263,278 jobs. Most jobseekers migrate to Selangor and Kuala Lumpur to find better jobs, as these two states witnessed a rapid expansion in Malaysia's development. Last but not least, in the eastern region, Terengganu was selected, as this state had the highest unemployment rate compared with other eastern states, at around 4.2% in 2016 [39].

### 3.3. Relative Importance Index

In order to understand the factors that contribute to unemployment, participants were asked to rank the reason for being unemployed according to their priorities. Respondents were presented with a set of alternatives and asked to order them from the most to the least preferred. In the ranking task, the alternatives were all presented at once in a randomised order, and respondents were asked to rank them. For instance, for $n = 5$ choice alternatives {A, B, C, D, E}, a ranking task involved filling the blanks with numbers from one (1), the most preferred factor, to five (5), which indicated the least preferred factor [40]. The method of ranking played a critical role in measuring preferences, attitudes, and values.

To determine the ranking factor, the Relative Importance Index (RII) was employed. According to [41], relative importance is the proportionate contribution each predictor makes to $R^2$, considering both the unique contribution of each individual as a predictor and its incremental contribution when combined with another predictor. The formula is as follows:

$$\text{RII} = \text{Sum of weights } (W1 + W2 + W3 + \ldots \ldots + WN)/A \times N \qquad (1)$$

where:

W = weighting given to each factor by the respondents, which ranges from 1 to 7; 1 is low priority and 7 is an essential priority;

A = highest weight (i.e., 7 in this case);

N = total number of respondents.

The indexes were ranked.

The comparison of the RII with the corresponding importance level was measured from the transformation matrix as proposed by [42]. According to the authors, the derived importance levels from the RII are given in Table 4:

**Table 4.** Importance levels from Relative Importance Index (RII).

| | |
|---|---|
| High (H) | $0.8 \leq \text{RII} < 1.0$ |
| High-Medium (H–M) | $0.6 \leq \text{RII} < 0.8$ |
| Medium (M) | $0.4 \leq \text{RII} < 0.6$ |
| Medium-Low (M–L) | $0.2 \leq \text{RII} < 0.4$ |
| Low (L) | $0.0 \leq \text{RII} < 0.2$ |

Source: [42].

For the measurement of reliability, the study used Kendall's coefficient of concordance, which measures the degree of agreement among respondents. This coefficient basically produces a scale of 0 (no agreement) to 1 (perfect agreement), which is used to evaluate the level of ordinal scoring agreement [43]. Generally, values of 0.75 and above suggest an "excellent" level of agreement, values of 0.40–0.75 indicate "fair–good" agreement, and values of 0.40 and below represent "poor" levels of agreement or agreement which is almost random [44].

These questions provide a small but potentially important insight into individual preferences regarding the role of each variable when it comes to the decision to enter the labour market.

*3.4. Questionnaire Pretesting*

In the pretesting phase, prior information was obtained from various sources, experts, and jobseekers in order to comprehensively take into account all the relevant problems of the study. A total of 10 jobseekers were asked to respond to the questionnaire to ensure the clarity of the questions. The gathered feedback pertains to the reasonableness of the bid amount, confirmation of the list of factors, and clarity of the questions. The pretesting survey was conducted in February 2017.

*3.5. Pilot Test*

Pilot testing is an important step to improve the quality of a questionnaire and determine the problems in completing it. The pilot test for this study was conducted in February 2017 with a total of 40 respondents. Feedback from the pilot test was then gathered.

**4. Results and Discussion**

*4.1. Descriptive Statistics*

Table 5 presents the descriptive statistics of the respondents. Single and female respondents were slightly overrepresented in the sample with 82% and 68%, respectively. Female overrepresentation might be because the number of unemployed females is relatively higher than that of males. The unemployment rate for females was only 3.1% in 2000, and it rose significantly to 3.9% in 2016, while the unemployment rate for males was 3.0% in 2000, yet this rate remained stable until 2016 [45]. Regarding age, 45% of respondents were aged 20–24, 44% were 25–29, and 11% were 30 and above. In terms of the location of respondents, 59% of them came from urban locations, while the rest were from rural areas, with 21% of them from the south of Peninsular Malaysia, 25% from the north of

Peninsular Malaysia, 25% from the west of Peninsular Malaysia, and 29% from the east of Peninsular Malaysia.

**Table 5.** Characteristics of respondents.

| Education Level | | Diploma | Bachelor's Degree | Master's Degree and Ph.D. |
|---|---|---|---|---|
| Gender | Male | 56 | 66 | 7 |
| | Female | 76 | 183 | 14 |
| Status | Single | 102 | 213 | 17 |
| | Married | 30 | 36 | 4 |
| Age | 20–24 | 75 | 106 | 2 |
| | 25–29 | 39 | 124 | 15 |
| | 30–34 | 11 | 15 | 3 |
| | 35–39 | 4 | 3 | 1 |
| | 40–44 | 3 | 0 | 0 |
| | 45–49 | 0 | 1 | 0 |
| Zone | South | 25 | 53 | 5 |
| | North | 23 | 65 | 11 |
| | West | 50 | 49 | 3 |
| | East | 34 | 82 | 2 |
| Strata | Urban | 87 | 134 | 16 |
| | Rural | 45 | 115 | 5 |
| Specialisation | Social Science | 81 | 152 | 15 |
| | Science | 51 | 97 | 6 |
| University | Public University | 94 | 219 | 21 |
| | Private University | 38 | 30 | 0 |
| Year of graduation | 2017–2014 | 88 | 207 | 19 |
| | 2013–2010 | 28 | 31 | 1 |
| | 2009–2006 | 8 | 9 | 1 |
| | 2005–2002 | 4 | 2 | 0 |
| | 2001 and earlier | 4 | 0 | 0 |
| Duration looking for a job | Less than 3 months | 62 | 136 | 10 |
| | 3 months to less than 6 months | 29 | 45 | 8 |
| | 6 months–less than a year | 21 | 33 | 2 |
| | 1–3 years | 14 | 27 | 1 |
| | More than 3 years | 6 | 8 | 0 |
| Frequency of looking for a job | Almost every day | 58 | 158 | 12 |
| | Almost every week | 43 | 63 | 6 |
| | Almost every month | 8 | 8 | 1 |
| | Uncertain | 23 | 20 | 2 |
| Total of respondents | | 132 | 249 | 21 |

Regarding the education level of respondents, 33% held a diploma, 62% of them held a bachelor's degree, and 5% held a master's degree and above. This proportion could reflect the real situation of jobseekers. The data from the Department of Statistics, Malaysia

(DOSM), revealed that in 2016, the number of unemployed diploma holders had increased threefold compared to the previous six years, while the number of unemployed degree holders was five times higher. In terms of educational background, 62% were from the social science field while the rest were from science and engineering fields. Most were young graduates since 78% of them graduated between 2014 and 2017, while another 15% graduated between 2010 and 2013, and the rest graduated in or before 2009.

Furthermore, 83% of the respondents graduated from public universities. Even though a study conducted by [46] highlighted that employers prefer to hire employees who graduated from public universities, compared to private universities, graduates from public universities are still struggling to find jobs. The study [47] also stressed that factors such as reputations of institutions affect the decisions of employers to hire fresh graduates. The descriptive also shows that around 70% of them had been eagerly looking for a job over the past 6 months and looking for a job almost every day. This circumstance indicates the eagerness of young graduates to look for jobs and be actively involved in the job-hunting process.

In terms of the channels used to find a job, [48,49] highlighted that jobseekers could search for jobs through formal (newspapers, employment agencies, online) and informal channels (friends' or relatives' networks). The present survey shows that respondents use the Labour Department, employment agencies, and job fairs as their main channels to find jobs (Table 6). They believe that these channels are the most effective to match their needs with the job opportunities in the market. This finding is also consistent with a study in [50], which found that jobseekers would go to public employment services and job fairs or open interviews when they are prepared to go into the labour market.

**Table 6.** Channel to find jobs.

| Channel to Find Jobs | RII | Rank | Importance Level |
| --- | --- | --- | --- |
| Labour Department/Employment Agency/Job Fair | 0.83 | 1 | H |
| Send applications | 0.72 | 2 | H–M |
| Answer advertisement/advertise (online/company website) | 0.71 | 3 | H–M |
| Inform friends/relatives | 0.50 | 4 | M |
| Others | 0.24 | 5 | M–L |

However, in terms of hiring practices, the survey found that employers prefer to use online advertisements or informal networks. This suggests that jobseekers and employers use different channels, distorting the matching process. Much empirical evidence suggests that about half of all jobs are filled through contacts or networks [48]. Employers prefer to use their social networks to fill vacancies and it could produce good matches at a higher rate than formal methods, thus potentially reducing mismatches [49]. This is also supported by other studies that suggested nearly 30–60% of new hires are recruited through informal methods [51–53].

### 4.2. Factor of Mismatches

Understanding the determinants of being unemployed could shed light on various aspects of job search behaviour. The results show that jobseekers' belief that no suitable jobs are currently available in the market is the most common reason for being unemployed, based on the computing of Relative Importance Index (RII) values as shown in Table 7. In terms of general concordance from all respondents, the rate of agreement among respondents based on the sample gathered is 0.68, which indicates fair to good agreement between the respondents.

**Table 7.** Relative Importance Index (RII) for reasons for being unemployed.

| Factor | RII | Rank | Importance Level |
|---|---|---|---|
| Believe no jobs available | 0.82 | 1 | H |
| Wages | 0.72 | 2 | H–M |
| Location mismatch | 0.70 | 3 | H–M |
| Educational mismatch | 0.67 | 4 | H–M |
| Soft skills mismatch | 0.52 | 5 | M |
| Family Responsibilities | 0.44 | 6 | M |
| Others | 0.23 | 7 | M-L |

As jobseekers ranked the factor "believe no jobs available" as the most important reason, with 0.82 (RII), it suggests that they lack knowledge about the jobs available in the market. This is due to the higher information asymmetry on the labour market, which resulted from vacancies unreported to jobseekers, those which failed to reach them. In the case of Malaysia and most other countries, it is not an obligation to report vacancies to the responsible agency; thus, this circumstance affects the decisions of jobseekers to enter the labour market. As mentioned by [54], a lack of information about a vacancy will eliminate the job from the job search. The study [55] also stated that the availability of appropriate information for the labour market is among the most important requirements for effective labour matching and a prominent part of active labour market policies.

This result could give a new insight into the problem of graduate unemployment. As mentioned by [18], the real problem is not because of the limited employment opportunities, but because graduates are not work-ready. However, when jobseekers do not even have information regarding career opportunities, the supply side cannot take the blame alone. This is also supported by data from the Ministry of Human Resources (MOHR), which showed that the creation of job vacancies these days is mostly concentrated on medium- and low-skilled jobs which do not match the job expectations of graduates.

The study [56] mentions that among the reasons for being self-employed are job satisfaction and the lack of available alternatives. As the landscape of the labour market is changing due some shocks to the economy, such as the occurrence of the COVID-19 pandemic, many young graduates venture into the informal sector. With the expansion of the gig economy, the number of jobs in the informal sector, such as freelancers and food riders (working for companies such as Grab Food and Food panda), is growing remarkably. As people are advised to stay at home, there has been an increase in demand for food distribution services in South East Asia, which creates opportunities for food riders.

In Malaysia, it has been stated that some distribution firms have seen a rise in food orders of more than 30% since the Movement Control Order (MCO) was implemented on 18 March 2020 [57]. Even though gig work offers flexible job opportunities, other concerns are arising, such as limited access to social protection, job insecurity, and unstable incomes. In the case of Malaysia, data from the DOSM state that 9.4% of total employment in 2017 comprised workers in the informal sector. Almost two-thirds of the employment in this sector was concentrated in the services industry (62.1%), followed by construction (20.0%). Besides, 17.2% worked in manufacturing and 0.7% in other fields and agriculture.

Wages was another reason for unemployment among the respondents. Since they pursued their studies to tertiary level, they expect a more rewarding wage as promised by human capital theory. Even though [58] mentioned that bachelor's degree holders have a higher average return compared to others, [59] revealed that half of all graduates (53.7%) make less than MYR 2000 per month. Usually, low wages also reflect fewer opportunities for career advancement and fewer benefits from the opportunities created [54]. Also, it indicates that either low skill or low education levels are required by the employer, or that the employer is unwilling to pay more for the expected skill. Nevertheless, this wage

penalty might affect employees in the early stages of their careers and be a temporary phenomenon, as the process of finding a suitable job may take some time [60].

Even though the implementation of a minimum wage has supported increases in the salaries of lower-skilled workers in recent years, and the real starting salaries of Penilaian Menengah Rendah (PMR) and Sijil Pelajaran Malaysia (SPM) or equivalent to Senior Cambridge Certificate, GCE O Level-educated employees have risen by 4.6% and 2.3%, respectively, the starting salaries of graduates have declined in real terms. A real starting monthly salaries for most fresh graduates have deteriorated since 2010. A fresh graduate with a diploma earned a real salary of only MYR 1376 in 2018, which had decreased by 6% compared to MYR 1458 in 2010. While master's degree holders earned a real salary of MYR 2707, this is a significant decline from the equivalent in 2010 of MYR 2923, after adjusting for inflation [61]. This divergence in growth trends across education levels introduces a more serious phenomenon: the income premium for education has lessened in Malaysia. If left unaddressed, this could diminish the efforts of the younger population to pursue tertiary education and could cause the "brain drain" issue in Malaysia to deteriorate.

Geographical mismatch is also another prime generator of unemployment, as a lack of access to job vacancies causes a reduction in search efforts [62]. According to the MOHR and Bank Negara Malaysia (BNM), most job vacancies are in Selangor (734,163 vacancies), followed by Sabah, Johor, Kuala Lumpur, and Penang. In this case, jobseekers from other parts of Malaysia need to commute to find better jobs and opportunities as the rate at which jobseekers find jobs depends on several factors, such as the rate of willingness to move to the place with available jobs. Other factors such as the rate of jobs opening in one location and the rate at which employed workers vacate jobs in a location that matches the jobseekers' location also contributed to this [63].

On the other hand, when jobseekers were asked the main element for considering a job, they chose qualification, wages, self-satisfaction and location as the important elements, as shown in Table 8. It can be suggested that in this current labour market, jobseekers believe no suitable vacancies are available based on their qualifications, expected wages, self-satisfaction and location. The creation of job vacancies that are concentrated on low- and medium-skilled work coupled, and uneven distribution of vacancies across the region with asymmetric information are contributing to this problem.

**Table 8.** Relative Importance Index (RII) for priorities when considering a job offer.

| Factor | RII | Rank | Importance Level |
| --- | --- | --- | --- |
| Qualification | 0.82 | 1 | H |
| Wages | 0.81 | 2 | H |
| Self-satisfaction | 0.80 | 3 | H |
| Location | 0.80 | 4 | H |
| Family demand | 0.66 | 5 | H–M |
| Organisational culture | 0.64 | 6 | H–M |
| Others | 0.47 | 7 | M |

*4.3. Job Preferences*

In descending order, respondents prefer to venture into the services sector, then manufacturing, construction, and agriculture/forestry/fishing/mining/quarrying (Table 9). This finding is also supported by a study from [55], which revealed that the services sector is the most preferred sector to work in, specifically the subsectors of education, civil service/public administration/uniform services, accommodation, and food and beverage service activities.

**Table 9.** Sector preferences.

| Sector | RII | Importance Level | Rank |
|---|---|---|---|
| Services | 0.74 | H–M | 1 |
| Manufacturing | 0.70 | H–M | 2 |
| Construction | 0.59 | M | 3 |
| Agriculture/forestry/fishing/mining/quarrying | 0.58 | M | 4 |
| Others | 0.38 | M–L | 5 |

The total vacancies in 2016 numbered 854,044. Data from BNM show that they were mostly distributed in the preferred sectors. The two sectors with the most vacancies were manufacturing (44%) and services (20%). They were followed by agriculture/forestry/fishing/mining/quarrying (21%) and construction (15%). However, looking at the distribution of vacancies by group of occupation reveals that 67% of vacancies are offered in elementary occupations and 17% for plant and machine operators and assemblers. These are low- and medium-skilled groups which do not match the qualifications of graduates.

However, interestingly, even though the respondents have their own preference of sector, 78% of the respondents do not mind working in sectors that are not related to their field of specialisation and educational background (Figure 3). Some respondents do not mind working outside of their preferred sectors as long as the wages and salaries are acceptable and allow them to pay their monthly commitments. Some are not too selective, as they might find it rewarding to gain new knowledge and skills if they worked in non-specialised sectors. By working in a different environment, they could accumulate their human capital to gain deeper and broader experience and skillsets.

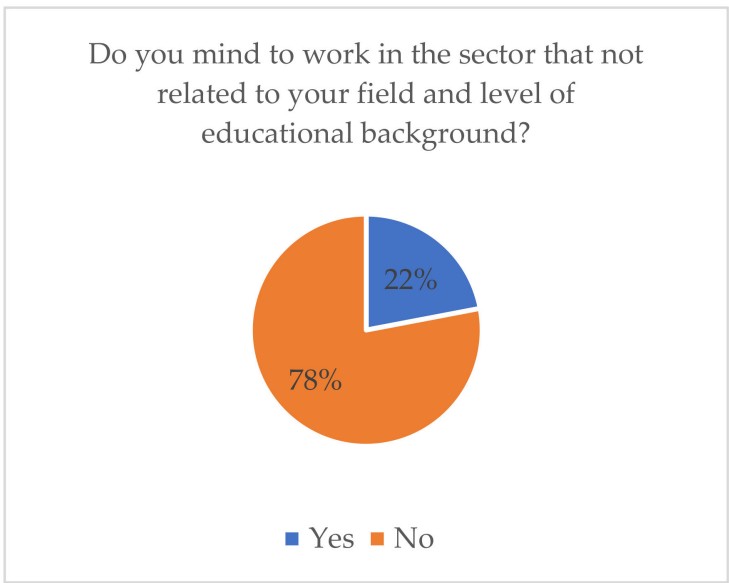

**Figure 3.** Job preferences.

## 5. Conclusions

The purpose of this research was to identify the factors behind unemployment from the supply side, specifically job seekers, as their preferences are very important when analysing the employment decision process. The findings have revealed that in general, job seekers believe that no suitable jobs are available in their respective market, which has happened because of the asymmetric information between the supply and demand sides

of the market. This circumstance has at least provided a rough proxy of information about job vacancies and unavailability in the market.

On the other side, there is a possibility of the incidence of mismatch in the labour market, as job seekers cited that the criteria of job vacancies are not met with their qualifications. The demands of the market are among the most important reasons for their unemployment. Although the government has provided various initiatives such as 66 million assistances for e-healing, if the problem of mismatch is not addressed, then the unemployment rate will still continue to inflate (increase) and challenge the validity of the theory of human capital.

Some solutions could be suggested—for example, employers could enhance the hiring process by posting job vacancy advertisements to make vacancies more visible to job seekers. Although Malaysia currently has "JobsMalaysia" as the main platform for finding jobs, the role of responsive agencies needs to be streamlined. The government has to enforce regulation on the agencies that are looking to hire to ensure they report all job vacancies, so that the responsible authority could have a better database for the country. From this practice, a strong database that consists of more detailed information about both employers and job seekers could be developed. In addition, employers should avoid the practice of posting false advertisements, a situation where firms do not have the intention to hire in the first place but still continue to post job vacancies in order to fulfil the condition or requirement of the hiring process, which could distort the matching process and its efficiency in the market.

Moreover, to expedite efficient matches between job seekers and employment agencies, active labour market policies should take priority over passive policies. Active policies that aim to increase the probability of unemployed workers finding jobs through more direct approaches—such as job creation, job sharing, job rehabilitation, training and job search assistance for unemployed workers, and incentivising employers to expand their workforces—could increase the matching process more efficiently.

For instance, efforts by the government such as Skim Latihan 1 Malaysia (SL1M), currently branded as PROTÉGÉ, are designed to equip Malaysian graduates with necessary skills and experience. It is one of many important efforts in preparing job seekers to embrace the labour market challenge, and it is worthy of its praise. To assist job seekers, especially fresh graduates, during this pandemic, the Ministry of Higher Education has also drawn up a strategy in order to assist new graduates. The Ministry has established the Graduates Reference Hub for Employment and Training (GREaT) to provide resources such as matching, re-skill and up-skill schemes, as well as grants for further education.

The Ministry's career advancement programme has also allocated MYR 100 million from the National Economic Recovery Plan (Penjana). It is roughly estimated that MYR 35 billion has been allocated to 40 Penjana programmes to help Malaysia's economy rebound from the COVID-19 pandemic. The Ministry's goal is to support at least 20,000 graduates under the plan, which comprises 140 individual services conducted in conjunction with 100 different companies. Meanwhile, to avoid major retrenchments during the pandemic, the government has also introduced a subsidy program where employers will receive subsidies for their workers for a period of up to six months, ranging from MYR 600 to 1200, based on the size of the workload.

In addition, there is an urgent need to induce a demand of quality labour through the creation of high-skilled jobs that match the excessive supply of graduates with tertiary education. In this regard, it is vital to attract new, quality investments from both foreign and domestic firms, pivoting away from the low-cost business model. Among existing firms, this can be generated through automation and moving up the value chain, with higher reliance on knowledge and technology, which could create high-skilled jobs. Doing so requires coherent investment policies, which likely involves reviewing and enhancing existing investment incentives.

This research is constrained by some limitations. As the study is focusing on the factors of unemployment from the supply side at a national level, it is essential to also look

at the factors specifically at the regional level, as each region might have different factors of unemployment due to the different nature of economic activities.

**Author Contributions:** Conceptualization, R.S. and S.J.; formal analysis, R.S. and S.J.; methodology, R.S. and S.J.; project administration, R.S, N.W.I. and N.M.N.; supervision, R.S., N.W.I. and N.M.N.; validation, R.S., N.W.I. and N.M.N.; writing—original draft, R.S. and S.J.; writing—review and editing, R.S., N.W.I. and N.M.N. All authors have read and agreed to the published version of the manuscript.

**Funding:** This research was funded by the Ministry of Education, Malaysia, under the Fundamental Research Grant Scheme (FRGS) (Project Code: FRGS/1/2015/SS08/UPM/02/6).

**Institutional Review Board Statement:** Not applicable.

**Informed Consent Statement:** Not applicable.

**Data Availability Statement:** Data available on request due to restrictions.

**Acknowledgments:** We would like to thank the anonymous reviewers for their valuable comments that helped to improve this manuscript considerably.

**Conflicts of Interest:** The authors declare no conflict of interests.

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
