# Peer review of "Are Jobs Available in the Market? A Perspective from the Supply Side"

_sustainability, doi:10.3390/su13041973_

Round 1
Reviewer 1 Report
The paper is interesting, but the conclusions are too general. First of all, the authors did not answer the question: "Are jobs available in the market?". The research conducted suggests that there are vacancies for graduates, but the possibilities to fill them are low due to mishmash and imperfect information flow. Thus, the paper postulates the need to improve the flow of information between the supply and demand sides on the labour market. It is true, but it is not enough in order to close the mismatch between the characteristics of graduates and vacancies. In my opinion, the weakness of the paper is the over-generalization of the situation of graduates on the labour market. It is an internally diversified group, just like the demand for labour. Thus, there is no single level of equilibrium (structural) unemployment for all graduates, and there are no universal solutions that are good for all groups of graduates. Identification of partial equilibria would make it possible to fine-tune the remedial measures and at least approximate how much the unemployment level of individual groups is sensitive to structural and cyclical factors. Perhaps it will be difficult to show this problem based on the empirical data in this paper, but at least it should be mentioned in the course of the analysis (e.g. based on literature, not necessarily related to Malaysia). As far as some references to literature are concerned, for a better readability, apart from the number of a given item, it is also worth mentioning the author's name (see e.g., lines 131, 134, 139).
Reviewer 2 Report
Type of manuscript: Empirical article
Title: Are jobs available in the market? A perspective from the supply side
Comments:
The present study investigates the reasons behind the long-term unemployment issue from the supply side. Using the Relative Importance Index (RII), this study investigated 402 respondents at selected job fairs to identify the cause of their difficulty entering the labour market. The findings revealed that the unemployed believe the principal cause of their unemployment is the lack of suitable jobs for them in the market. This circumstance sends a signal of asymmetric information between demand and supply in the labour market, especially to young graduates. The study has several merits and consists of a very innovative empirical work. However, after a further round of an in-depth review by the authors, I would like to comment on the following academic weaknesses that should be addressed before any consideration for publication.
Recommendation: Major revision
More specifically,
A major revision is requested before any attempt for publication will most likely require to take into account the following comments.
- Introduction is not well written. It is too long with too much information that should be present in another section (for example the situation in…). In the introduction part, the main aim and the main contribution/innovation of the study should be very clear. Here it is not presented.
- Authors should create a new subsection after the introduction in which literature review should be added. Literature review is plenty in this field and it will give more assets to the paper.
- Regarding the empirical investigation, theoretical framework is missing. Authors give a lot of descriptive information whereas empirical results are not presented very well. Comparison with previous empirical studies is needed.
- Conclusion part is nice. However, policy implications should be retrieved from the empirical estimations. They are very general. Also, limitations and further research should be very clear for the reader.
Round 2
Reviewer 2 Report
Dear authors,
After the second round review process, I can mention that you made all required changes. Now, I can see that the quality of your article has been highly improved.